# Beneficial Effects of Oral Nutritional Supplements on Body Composition and Biochemical Parameters in Women with Breast Cancer Undergoing Postoperative Chemotherapy: A Propensity Score Matching Analysis

**DOI:** 10.3390/nu13103549

**Published:** 2021-10-10

**Authors:** Joanna Grupińska, Magdalena Budzyń, Kalina Maćkowiak, Jacek Jakub Brzeziński, Witold Kycler, Ewa Leporowska, Bogna Gryszczyńska, Magdalena Paulina Kasprzak, Maria Iskra, Dorota Formanowicz

**Affiliations:** 1Chair and Department of Medical Chemistry and Laboratory Medicine, Poznan University of Medical Sciences, 60-806 Poznań, Poland; magdalenabudzyn@ump.edu.pl (M.B.); kmackowiak@ump.edu.pl (K.M.); bognagry@ump.edu.pl (B.G.); magdarut@ump.edu.pl (M.P.K.); iskra@ump.edu.pl (M.I.); doforman@ump.edu.pl (D.F.); 2Hospital Pharmacy, Greater Poland Cancer Centre, 61-866 Poznań, Poland; 3Department of Oncological Surgery of Gastrointestinal Diseases, Greater Poland Cancer Centre, 61-866 Poznań, Poland; jacek.brzezinski@wco.pl (J.J.B.); witold.kycler@wco.pl (W.K.); 4Department of Laboratory Diagnostics, Greater Poland Cancer Centre, 61-866 Poznań, Poland; ewa.leporowska@wco.pl

**Keywords:** oral nutritional supplements, chemotherapy, breast cancer, sarcopenic obesity, hypoalbuminemia

## Abstract

**Aim:** Recently, more attention has been paid to the role of nutritional intervention in preventing the side effects of chemotherapy in oncology patients. Therefore, the aim of the present study was to analyze the effects of oral nutritional supplements on the body composition and biochemical parameters in women with breast cancer receiving postoperative adjuvant chemotherapy. **Patients and Methods**: The study involved women diagnosed with breast cancer who underwent surgical treatment and were qualified for chemotherapy (doxorubicin and cyclophosphamide). Women were divided into two groups, depending on whether oral nutritional supplements were used during chemotherapy. Anthropometric and biochemical parameters were analyzed twice in all patients: before and after six weeks of chemotherapy. Propensity score (PS) matching was performed to select patients balanced in terms of age, BMI, and clinicopathological features of the tumor. Statistical comparisons were conducted in a propensity-matched cohort of patients. **Results:** The value of BMI was maintained constant in the supplemented women older than 56 years after six weeks of chemotherapy. Regardless of age in the supplemented women, a significant increase in muscle mass, fat free mass (FFM), and fat free mass index (FFMI) was demonstrated. An increase in fat mass (FM) including visceral fat was observed only in the non-supplemented control. Regardless of age or initial FM, supplemented women exhibited a constant level of albumin. Moreover, in the supplemented women with normal initial FM, the stable values of triglycerides and HDL cholesterol were maintained after six weeks of chemotherapy. **Conclusion:** The present study demonstrated that oral nutritional supplements could improve body composition and prevent hypoalbuminemia and lipid abnormalities in women with breast cancer undergoing chemotherapy.

## 1. Introduction

Breast cancer is the most common malignant tumor and one of the leading causes of cancer-related death in women [1,2]. It is well known that nutritional status influences both the risk of developing breast cancer and anticancer treatment outcomes [3,4,5]. Oncology patients have complex nutritional problems, depending on the location and stage of cancer, which can further exacerbate during cancer treatments [6,7]. Chemotherapy is the standard procedure for treating women with breast cancer. Although this treatment increases the chance of survival, it causes many adverse effects, resulting in inadequate nutritional status. Certain food groups are often rejected or preferred during chemotherapy [8,9] because of side effects of treatment such as nausea and vomiting [10]. Chemotherapy per se and accompanying inadequate nutritional status can drastically affect body composition and biochemical parameters, mainly albumin and lipid level. This is manifested by disturbed body mass index (BMI), fat mass (FM), muscle mass content, altered lipid profile, and hypoalbuminemia in patients receiving chemotherapy [11,12,13].

For women with breast cancer, changes in dietary patterns during chemotherapy are not always accompanied by weight reduction, since 50–96% of women with breast cancer have weight gain during treatment, with progressive gains in the months and years following the diagnosis [14,15]. However, weight gain does not have to be a sign of good nutrition but rather a diet rich in fat and sugar. The initial excess weight, or that acquired during the development of the disease, is a factor that negatively influences the prognosis, quality of life, and survival of women affected by breast cancer [16]. In addition, women receiving chemotherapy are exposed to adverse changes in body composition, such as frequent sarcopenia (loss of muscle mass), which is accompanied by an increase in FM [17]. These changes are associated with poor clinical outcomes, including lower disease-free and progression-free survival and higher overall mortality [18,19,20,21]. The immediate rise in total cholesterol, LDL cholesterol, and triglycerides levels during chemotherapy in patients with breast cancer has almost become an indisputable fact [12,22,23]. Recent studies have indicated that dyslipidemia does not diminish rapidly after chemotherapy [23]. Persistent dyslipidemia among patients with breast cancer may significantly increase the risk for the development of comorbidities and long-term survival [23]. In addition, patients receiving chemotherapy are exposed to a decrease in albumin, which is associated with a higher incidence of severe symptoms of chemotherapy-induced toxicity [24].

There is a high probability that maintaining balanced nutrition during chemotherapy could minimize the anthropometric and biochemical changes described above. As a result, the effectiveness of treatment, survival, and quality of life may be significantly improved. Evidence for this is provided by studies showing that nutritional intervention, including dietary counseling, oral nutritional supplements (ONS), and enteral nutrition, positively affects the nutritional and clinical status outcomes of patients receiving chemotherapy [25]. Particular attention has been paid to the role of ONS in reducing the adverse effects of chemotherapy by maintaining a well-balanced diet. ONS is a ready-made product that contains balanced nutrients, calories, and proteins to complement insufficient oral intake [26,27]. Most of the current research on the use of ONS has focused on esophageal cancer, gastrointestinal cancer, and head and neck cancer [28,29,30]. In these patients, ONS allows maintaining a stable weight, reduces skeletal muscle loss and sarcopenia prevalence, and improves chemotherapy tolerance [28,29,30]. However, limited studies have investigated the clinical effects of nutritional interventions on patients with breast cancer receiving chemotherapy. Therefore, this study aimed to assess the impact of ONS on anthropometric and biochemical parameters, including albumin and lipid concentration in patients diagnosed with breast cancer and treated by adjuvant chemotherapy.

## 2. Materials and Methods

### 2.1. Study Participants

The study involved patients of the Greater Poland Cancer Center in Poznań diagnosed with breast cancer. All women underwent surgical treatment and were qualified for adjuvant chemotherapy (doxorubicin and cyclophosphamide). The study’s exclusion criteria were the coexistence of diseases such as hyperthyroidism and hypothyroidism, Cushing’s disease, adrenal medulla disease, diabetes, chronic kidney and liver disease, and autoimmune diseases. Patients with a history of other cancer or recurrent breast cancer in the last five years were also excluded. During the first interview preceding chemotherapy, the clinician informed each patient about the benefits of oral nutritional intervention and recommended commercially available nutritional drinks. Each patient gained knowledge about the properties of the recommended nutritional drink and its proper oral administration. The patients made a voluntary decision about supplementation. Out of the ninety-eight patients recruited for the study, thirty-eight of them decided to take oral nutritional supplements, and sixty did not express their will. The study was performed according to the principles of the Declaration of Helsinki [31]. The investigational protocol was approved by the Local Bioethical Committee of Poznan University of Medical Sciences (no. 245/2015 5 March 2015). Written informed consent was obtained from all subjects in the study. The study scheme is presented in Figure 1.

### 2.2. Oral Nutritional Supplements (ONS)

Patients assigned to the supplemented group were required to receive 125 mL of oral nutritional supplements (Nutridrink protein^®^, 300 kcal, 18 g of protein per 125 mL, Nutricia Poland) two times a day after breakfast and early dinner (until 6 p.m.). Patients not taking nutritional supplements were assigned to the non-supplemented control group. To evaluate the effect of the supplementation depending on the age of the patients and their initial FM, supplemented women were divided into subgroups according to the FM (≤33% normal, >33% excessive) and age (≤56 years and >56 years). The cut-off point for FM was set at 33%, according to the guidelines of the WHO Expert Committee, which in 2004 found that overweight corresponds to 31–39% body fat in females [32]. The age of 56 years was taken as the cut-off point because it was the median age observed in the studied population of supplemented women.

### 2.3. Anthropometric and Biochemical Parameters Measurements

All participants underwent anthropometric and biochemical examinations twice: on the first day of chemotherapy, before drug administration, and six weeks later when the third cycle of chemotherapy was started. The anthropometric analysis was performed using the electric bioimpedance method (BIA) by a certified, 8-electrode Tanita BC 418-MA body composition analyzer. Body weight, fat mass (FM), muscle tissue content, fat free mass (FFM), water content, basal metabolic rate (BMR), and visceral tissue were assessed for all participants. Based on the obtained data, body mass index (BMI), waist to hip ratio (WHR), and fat free mass index (FFMI) were calculated. Biochemical parameters (albumin, transferrin, bilirubin, creatinine, urea, triglycerides (TAG), HDL cholesterol, glucose, AlAT, AspAT, and gamma-glutamyl transferase (GGT) were assessed in the Laboratory Diagnostics Department of the Greater Poland Cancer Center. The material for biochemical determinations was a blood sample taken by the technician in the morning hours from the forearm vein of the participants. Fasting blood was drawn from patients in a recumbent position by the same technician. Blood was collected in plasma tubes with heparin or EDTA as the anticoagulant, respectively. After 30 min, the tubes were centrifuged at 3000 rpm for 15 min to obtain plasma. The biochemical parameters were determined immediately using COBAS 6000/c501 apparatus (Roche/Hitachi). For this purpose, reagents from the same producers were always used, and the same temperature of determinations was kept. Before each measurement, the apparatus was calibrated, which guaranteed the appropriate repeatability and reproducibility of the results.

### 2.4. Statistical Method

Statistical analyses were performed using Statistica version 12.0 (StatSoft Inc., Tulsa, OK, USA). Two independent sample *t*-tests and chi-square tests or Fisher’s exact tests were used to assess baseline data between the control and the supplemented group. The propensity score (PS) matching method was used to balance the differences in several baseline characteristics between the two groups (Table 1). A total of seven characteristics (Table 1) were assessed for inclusion in the model as independent variables. All seven characteristics were retained in the model with stepwise selection and were subsequently used to generate propensity scores. In the selection process, a *p*-value of 0.05 was used as a cut-off for a characteristic to be entered and remain in the model. The baseline characteristics were assessed by evaluating the standardized mean differences; a standardized mean difference numerically <0.1 was considered acceptable. A cohort of seventy-six matched patients was ultimately achieved, with the same number in the control (*n =* 38) and the supplemented group (*n =* 38) balanced in terms of age, BMI, and clinicopathological features of the tumor. The baseline characteristics of the overall and propensity-matched cohort of patients are given in Table 1.

After PS matching, the matched control and supplemented group were subjected to statistical comparisons. The normality of quantitative variables was tested using the Kolmogorov–Smirnov or Shapiro–Wilk test. Normally distributed, continuous variables were presented as a mean and standard deviation. A comparison between analyzed parameters before and after chemotherapy was performed using a paired Student’s *t*-test. To compare the magnitude of changes of the analyzed parameters between the groups, the delta values (the difference between the initial and final value) were calculated and compared using an unpaired Student’s *t*-test or a Mann–Whitney U-test depending on the distribution of the data. In all cases, a *p*-value ≤ 0.05 was considered significant.

## 3. Results

### 3.1. Baseline Characteristics of Patients

We enrolled ninety-eight women who met our inclusion criteria, thirty-eight of whom underwent ONS during chemotherapy. Before PS matching, there were differences in several baseline characteristics between the supplemented group and the non-supplemented control (Table 1). We found that women from the control group were older and had a significantly lower value of BMI and smaller tumor size. Under PS matching, thirty-eight supplemented women were well matched to thirty-eight women from the control. The variables entered were very similar and comparable between the supplemented group and the control. Several variables, including age, BMI, and tumor size, became more comparable after PS matching.

### 3.2. The Effect of ONS on Anthropometric and Biochemical Parameters after Six Weeks of Chemotherapy

The measurements of anthropometric parameters were performed twice: before chemotherapy and six weeks later in the control and supplemented group. Based on the obtained results, an increase in body mass and BMI were observed in both groups (Table 2). The delta analysis showed that the BMI value changes to a similar extent in the control and the supplemented group (Table 3). In contrast to the control group, in the supplemented group, significant increases in muscle mass, FFM, and FFMI were observed (Table 2). Moreover, a rise in FM including visceral fat was observed in the control, whereas in the supplemented group, these parameters showed a statistically non-significant tendency to decrease (Table 2). The delta analysis confirmed that FM, including visceral fat, increased to a greater extent following chemotherapy in the control group than in the supplemented group (Table 3). Other anthropometric parameters remained constant in both the control and the supplemented group. However, only the water content increases to a greater extent following chemotherapy in the supplemented group than in the control (Table 3).

Similar to anthropometric parameters, biochemical markers such as albumin and transferrin, parameters reflecting kidney and liver function (creatinine, urea, GGT, AlAT, AspAT, bilirubin), lipids (TAG, HDL-cholesterol), and glucose were determined twice: before chemotherapy and six weeks later in the control group and the supplemented group. Contrary to the control group, the albumin concentration in the supplemented group remained constant after six weeks of chemotherapy (Table 2). The delta analysis revealed that albumin levels decrease to a greater extent following chemotherapy in the control than in the supplemented group (Table 3).

A fall in transferrin and bilirubin level, a rise in the activity of AspAT and GGT, and an increase in the concentration of TAG with a simultaneous decrease in the concentration of HDL cholesterol were noticed in both groups (Table 2). The delta analysis for these parameters showed a comparable magnitude of changes in both groups (Table 3). Urea, AlAT, and glucose concentration remained constant in supplemented group and control after six weeks of chemotherapy (Table 2). Additionally, to investigate the impact of factors other than supplementation on the observed changes in biochemical and anthropometric parameters, we conducted a multivariate regression analysis in which an independent influence of age and the clinicopathological features of the tumor were examined (Table 4). The analysis showed an increased age of the patients as an independent factor contributing to the reduction of HDL cholesterol concentration. In the case of other changes observed during chemotherapy in the supplemented group, neither the age nor the clinicopathological features of the tumor seem to have any effect (Table 4).

### 3.3. The Influence of Initial FM on Biochemical Parameters in the Matched Supplemented Group after Six Weeks of Chemotherapy

Regardless of initial FM in the supplemented group, a constant albumin concentration was observed after six weeks of chemotherapy (Table 5). The delta analysis for albumin showed no statistically significant differences between patients with normal and abnormal FM (Table 6). Regardless of FM, a decrease in the concentration of transferrin, bilirubin, and an increase in GGT and AspAT activity were observed (Table 5). Delta values for these parameters were comparable in subjects with normal and abnormal FM (Table 6).

In supplemented patients with abnormal FM, an increase in TAG and a fall in HDL cholesterol levels were observed. In contrast, these parameters remained constant in patients with normal FM (Table 5). The delta analysis confirmed that TAG and HDL cholesterol levels increase and decrease, respectively, to a greater extent following chemotherapy in women with abnormal FM (Table 6). Moreover, a decrease in creatinine concentration was observed in patients with abnormal FM, while no such trend was shown in those with normal FM (Table 5). The delta analysis also confirmed that creatinine level falls to a greater extent in patients with abnormal FM (Table 6).

### 3.4. The Influence of Age on Body Composition and Biochemical Parameters in the Matched Supplemented Group after Six Weeks of Chemotherapy

Regardless of the age of the patients, increases in the value of the muscle mass, FFM, and FFMI were observed (Table 7). Moreover, it was shown that these parameters increase to a similar extent in both age groups (Table 8). In patients aged 56 years or less, an increase in body mass and BMI values was demonstrated, whereas in older patients (>56 years), these parameters remained constant after six weeks of chemotherapy (Table 7). The deltas analysis showed that BMI increases to a greater extent in women aged 56 years or less (Table 8). In the case of other anthropometric parameters, their values remained constant in both age groups after six weeks of chemotherapy (Table 7). Regardless of the age of the patients, albumin levels remained unchanged after six weeks of chemotherapy (Table 7). In older women (>56 years), the transferrin concentration also remained constant, whereas in younger patients (≤56 years), its concentration decreased significantly after six weeks of chemotherapy (Table 7). Regardless of age, a rise in TAG level and a fall in HDL cholesterol, as well as a decrease in bilirubin concentration and an increase in AspAT activity, were demonstrated (Table 7). In women aged 56 years or less, an increase in GGT and AlAT activity was observed, which was not found in older patients (>56 years) (Table 7). However, only in patients older than 56 years, a statistically significant decrease in creatinine concentration was demonstrated (Table 7). Other biochemical parameters remained unchanged in both analyzed age groups (Table 7).

## 4. Discussion

One of the most commonly used treatments for breast cancer is neoadjuvant or adjuvant chemotherapy. It is well known that cytostatics administration decreases the secretion of host anabolic hormones and alters host metabolic response, resulting in abnormalities in protein, lipid, and carbohydrate metabolism [33]. These disturbances are manifested as changes in body composition and blood biochemical parameters. Their severity may be further aggravated by the poor nutritional status common in cancer patients. Malnutrition impairs tolerance to anticancer treatments and is associated with decreased response to treatment and quality of life as well as shorter survival time [34,35,36]. A growing number of studies demonstrate that in patients undergoing adjuvant chemotherapy, early dietary counseling, including oral nutritional intervention, improves body weight and albumin status as well as reduces the incidence and severity of toxicity, thereby avoiding treatment interruptions [37,38]. Meng et al. showed that early ONS administration in patients with advanced nasopharyngeal carcinoma results in a lower frequency of chemotherapy interruptions lasting more than three days [37]. Similar results were obtained in the study by Paccagnella et al. [38], who enrolled patients with head and neck cancer, in whom nutritional support was associated with lower delays in the administration of chemotherapy. The number of reports on the impact of nutritional intervention on the nutritional and clinical outcomes of patients with breast cancer is still limited. Therefore, our study aimed to evaluate the influence of high-protein nutritional support on body composition and biochemical parameters in women with breast cancer undergoing postoperative adjuvant chemotherapy.

The tendency for patients with breast cancer to gain weight when receiving adjuvant chemotherapy has been reported consistently over the past two decades [17,39,40]. Weight gain during breast cancer treatment was first documented in the 1970s [41]; since then, prospective studies have reported higher rates of weight gain for women treated with chemotherapy than with other treatments such as surgery alone or radiation therapy [42,43,44]. A combination of behavioral and physiological factors has been purported to contribute to the weight gain observed in women following breast cancer treatment [45]. Previous studies have demonstrated that the menopausal status of patients with breast cancer appears to be a factor that influences the degree of weight gain. A lesser weight gain is observed in older postmenopausal patients than in younger premenopausal women [44,46,47]. One explanation for the more prevalent and greater weight gain in younger patients may be the impact of chemotherapy on ovarian function [45].

We have found that ONS maintains a stable weight only in postmenopausal women older than 56 years who are less likely to gain weight, as shown in the studies mentioned above. Although an increase in body weight was observed in younger, supplemented women (≤56 years), it was associated with increased muscle tissue and lean body mass. This finding confirms earlier studies by other authors who showed that oral supplementation of patients with lung and colorectal cancer increases metabolically active lean body mass [48,49]. This aspect of nutritional support appears to be of significant clinical importance, since it is known that losing lean body mass is an unfavorable prognostic factor in cancer patients [20].

Several investigators have explained the loss of muscle mass (sarcopenia) in tumor hosts exposed to chemotherapeutics. Anticancer drugs (including cisplatin, irinotecan, adriamycin, and etoposide) were shown to cause muscle wasting directly via activation of the NF-κB pathway and independently of the commonly implicated ubiquitin–proteasome system or indirectly via the production of pro-inflammatory cytokines, such as IL-1β, IL-6, and TNF, or by inducing oxidative stress and tissue injury [50,51,52]. Loss of muscle mass by deregulating cytokines release can favor tumor progression and growth. There is evidence that decreased muscle mass causes reduced expression and the liberation of IL-6 and IL-15, which play a significant role in the redistribution and infiltration of natural killer cells involved in tumor cells elimination [53,54,55]. Low muscle mass may also be associated with insulin resistance, as muscles are the target site of insulin-mediated glucose action, which is an additional risk factor for developing treatment complications [56,57].

It has been documented that chemotherapy regimens such as doxorubicin and cyclophosphamide damage the heart and skeletal muscles by overproducing reactive oxygen species. For this reason, patients with low muscle mass seem to be more susceptible to experiencing cardiotoxicity while undergoing chemotherapy. Our research demonstrated that ONS prevents muscle mass loss and may thus counteract the above-described processes.

We have also found that chemotherapy without supplementation promotes a rise in FM, including visceral fat in women with breast cancer. Excessive FM stimulates the aromatase activity while inhibiting the secretion of sex hormone-binding globulin, which increases the concentration of free estradiol, demonstrating pro-tumor effects [18,58]. Moreover, the high FM stimulates insulin secretion and insulin-like growth factor 1 (IGF-1), which are involved in the growth of both normal and neoplastic breast epithelial cells [19,59]. A further complication of chemotherapy in many obese patients is the increase in FM associated with loss of muscle mass, which leads to sarcopenic obesity [17,39,60,61]. Recently, there have been many systematic reviews and meta-analyses of the relationship between sarcopenia and clinical outcomes in cancer patients [62,63,64]. There is evidence that sarcopenic obesity is linked with a greater incidence of chemotherapy toxicity [65,66,67]. This is probably related to the wrong dose of chemotherapeutic agents administrated to the patients with sarcopenic obesity. Body-surface area has been used as an index of metabolic mass for the purpose of scaling chemotherapy dose. It is hypothesized that in sarcopenic patients, large BSA will drive a higher absolute chemotherapy dose that may distribute within and be metabolized and cleared by a very depleted lean body mass (LBM), resulting in a higher incidence of toxicity [57]. Our study revealed that ONS implemented in women with breast cancer by maintaining constant FM and stimulating muscle mass could prevent sarcopenic obesity and thus protect against the incidence of chemotoxicity.

It is well known that the appropriate albumin concentration is crucial for the effectiveness of anticancer therapy by delivering the drug to the target site of action [68]. Lis et al. [69] observed that a low albumin concentration correlates with a shorter survival time in women with breast cancer, regardless of the stage of the disease. Chemotherapy treatment causes a drastic drop in albumin levels, as previous studies have shown [70,71,72]. Ozdemir et al. found that after 15 days of chemotherapy, the total albumin level decreases to 92% of the initial value in children with acute lymphoblastic leukemia. Our study revealed that ONS allows maintaining a constant plasma albumin level during chemotherapy [73]. Our findings confirmed the results of Wang et al., who used albumin administration before chemotherapy in non-small cell lung cancer patients [13]. The authors demonstrated that albumin administration protects against decreased plasma albumin levels during chemotherapy and prevents chemotherapy-induced toxicity symptoms. In contrast, patients without albumin administration developed hypoalbuminemia [13].

Our study confirmed abnormalities in the lipid profile, lowered HDL cholesterol, and elevated triglycerides in patients with breast cancer receiving chemotherapy, as reported previously by other authors [12,22]. The observed disturbances may result from the influence of cytostatics on the expression of enzymes involved in lipid metabolisms such as lipoprotein lipase and lecithin cholesterol acyltransferase. Dyslipidemia is an unfavorable prognostic factor [74,75], since it leads to the development of cardiovascular disease, which is the leading cause of death in people recovering from breast cancer [23]. Our study revealed that ONS maintains a constant triglyceride level and HDL cholesterol in women with normal initial adipose tissue content during chemotherapy. Therefore, it may suggest that nutritional intervention could prevent chemotherapy-accompanying dyslipidemia.

This study has several limitations. First, the number of participants in both groups was small. Therefore, these preliminary observations serve as a proof of concept and the basis for future clinical trials involving larger patient cohorts. Second, in our study, data about the duration of hormone therapy was not collected, which could affect the body composition and eating habits of the patients. Another disadvantage of the study is the lack of information on the total energy consumption and patients’ physical activity during chemotherapy, which may strongly affect body composition. We also did not have exact knowledge about the daily consumption of proteins and fats in particular groups, which could impact the concentration of albumin and lipid metabolism. These aspects should be carefully analyzed in the future. However, this requires the use of food diaries in which the composition of the meals consumed would be carefully recorded. Due to the fact that studies such as ours are extended to at least a few weeks, this may pose some problems. The correct filling of food diaries requires strong self-discipline of patients over a relatively long period of time; otherwise, there is a risk of errors affecting the interpretation of the results. Therefore, as we could not control the filling of a dietary diary by patients in the following weeks of chemotherapy, we did not include the nutritional data in our studies.

## 5. Conclusions

The present study demonstrates that ONS protects against loss of muscle mass content, hypoalbuminemia, and lipid abnormalities in women with breast cancer receiving adjuvant chemotherapy. These findings argue that the safety and efficacy of anticancer treatment can be enhanced through nutritional intervention as a part of the integrated care approach.

## Figures and Tables

**Figure 1 nutrients-13-03549-f001:**
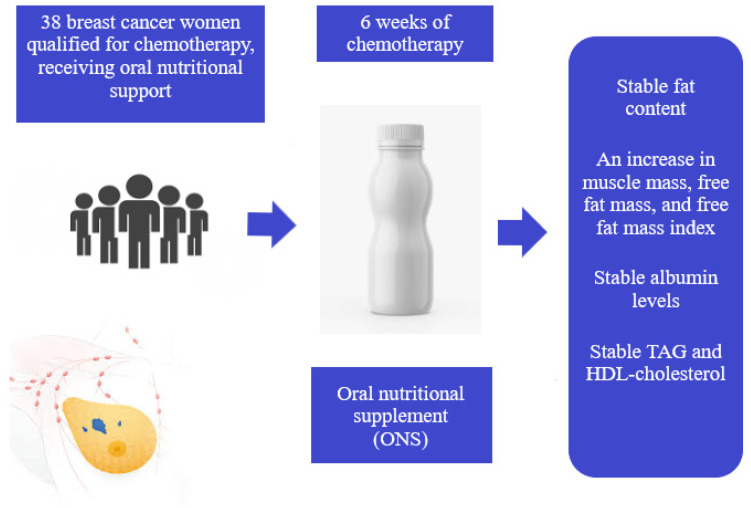
Scheme of the conducted research.

**Table 1 nutrients-13-03549-t001:** The baseline characteristics of the overall and matched cohort of patients.

		Overall Cohort (*n =* 98)						Matched Cohort (*n =* 76)			
		Supplemented Group (*n =* 38)		Control Group (*n =* 60)		P		Supplemented Group (*n =* 38)		Control Group (*n =* 38)	P
Age (mean ± SD)		55.42 ± 9.95		59.13 ± 7.97		0.0442 *		55.42 ± 9.95		56.47 ± 9.69	0.6416
BMI [kg/m^2^], (mean ± SD)		28.66 ± 6.50		26.30 ± 4.76		0.0403 *		28.66 ± 6.50		27.21 ± 5.16	0.2831
Clinicopathological features		N	%	N	%	P	N	%	N	%	P
Histopathological grade											
	I/II	21	55	31	52	0.7281	21	55	19	50	0.6459
	III	17	45	29	48		17	45	19	50	
HER-2/neu expression											
	(+)	13	34	20	33	0.9287	13	34	12	32	0.8071
	(−)	25	66	40	67		25	66	26	68	
Tumor size											
	<2 cm	18	47	41	68	0.0388 *	18	47	24	63	0.1663
	>2 cm	20	53	19	32		20	53	14	37	
Regional lymph node metastases											
	present	20	53	32	53	0.9459	20	53	16	42	0.3581
	absent	18	47	28	47		18	47	22	58	
Hormonal sensitivity											
	hormonal-positive	23	61	44	73	0.1841	23	61	27	71	0.3335
	hormonal-negative	15	39	16	27		15	39	11	29	

* statistically significant.

**Table 2 nutrients-13-03549-t002:** The effect of ONS on anthropometric and biochemical parameters after six weeks of chemotherapy in the matched supplemented group and control.

Analyzed Parameter	Matched Control Group (*n =* 38)			Matched Supplemented Group (*n =* 38)		
	before Chemotherapy	after 6-Week Chemotherapy	P	before Chemotherapy	after 6-Week Chemotherapy	P
Body mass [kg]	71.14 ± 12.7	72.32 ± 12.7	0.0009 *	77.14 ± 17.09	78.1 ± 17.31	0.0019 *
BMI [kg/m^2^]	27.21 ± 5.16	27.64 ± 5.09	0.0029 *	28.66 ± 6.5	29.03 ± 6.55	0.0015 *
FM [%]	32.69 ± 7.68	33.60 ± 7.02	0.0285 *	36.83 ± 8.2	36.2 ± 7.74	0.1976
WHR	0.87 ± 0.13	0.89 ± 0.19	0.5866	0.86 ± 0.08	0.87 ± 0.08	0.2117
Water content [%]	48.85 ± 5.62	48.4 ± 5.03	0.1996	46.48 ± 5.66	46.13 ± 8.82	0.1242
Muscle mass [kg]	44.28 ± 4.43	44.74 ± 4.85	0.0850	45.14 ± 5.23	46.23 ± 5.68	0.0020 *
Visceral fat	7.53 ± 3.04	7.76 ± 2.94	0.0481 *	8.66 ± 3.81	8.55 ± 3.7	0.3526
FFM [kg]	46.65 ± 4.69	47.1 ± 5.14	0.1057	47.56 ± 5.51	48.72 ± 5.96	0.0015 *
FFMI [kg/m^2^]	17.77 ± 1.88	17.93 ± 1.93	0.0882	17.71 ± 2.21	18.14 ± 2.36	0.0012 *
BMR [kcal]	1397.66 ± 146.63	1412.37 ± 156.96	0.0293 *	1546.13 ± 654.84	1475.68 ± 198.46	0.0002 *
Albumin [g/dL]	4.55 ± 0.25	4.41 ± 0.29	0.0045 *	4.51 ± 0.23	4.51 ± 0.27	0.9579
Transferrin [g/L]	2.65 ± 0.37	2.47 ± 0.38	0.0003	2.7 ± 0.4	2.54 ± 0.4	0.0004 *
Creatinine [mg/dL]	0.72 ± 0.11	0.69 ± 0.1	0.1529	0.77 ± 0.1	0.74 ± 0.08	0.0012 *
Urea [mg/dL]	30.27 ± 7.42	28.93 ± 8.24	0.2853	28.02 ± 6.93	28.05 ± 7.18	0.9279
GGT [U/L]	27.84 ± 21.57	42.55 ± 34.94	<0.0001 *	23.42 ± 14.81	33.18 ± 28.7	0.0003 *
Bilirubin [mg/dL]	0.46 ± 0.19	0.26 ± 0.12	<0.0001 *	0.44 ± 0.21	0.25 ± 0.1	<0.0001 *
Glucose [mg/dL]	104.74 ± 13.16	103.32 ± 13.88	0.7397	108.18 ± 22.62	105.58 ± 27.01	0.2615
TAG [mg/dL]	129.91 ± 78.06	174.18 ± 202.73	0.0088 *	127.97 ± 58.19	194.41 ± 136.17	0.0001 *
HDL-Cholesterol [mg/dL]	67.48 ± 16.68	61.87 ± 13.69	0.0014 *	60.03 ± 14.51	53.42 ± 14.54	<0.0001 *
AlAT [U/L]	19.87 ± 8.38	25.21 ± 14.86	0.0514	17.81 ± 9.83	20.61 ± 10.27	0.0070 *
AspAT [U/L]	18.68 ± 4.59	21.61 ± 8.07	0.0302 *	17.65 ± 6.01	20.05 ± 6.07	0.0006 *

All results shown as mean ± standard deviations. Paired Student’s test was used for comparison. * statistically significant.

**Table 3 nutrients-13-03549-t003:** Comparison of changes (before and after 6-week chemotherapy) for anthropometric and biochemical parameters between the matched supplemented group and control.

Analyzed Parameter	Matched Control Group	Matched Supplemented Group	P
	Δ	Δ_1_	
Body mass [kg]	1.17	0.96	0.5747
BMI [kg/m^2^]	0.43	0.36	0.5322
FM [%]	0.74	−0.63	0.0312 *
WHR	0.02	0.01	0.3857
Water content [%]	−0.45	0.62	0.0299 *
Muscle mass [kg]	0.45	1.09	0.1178
Visceral fat	0.24	−0.11	0.0263 *
FFM [kg]	0.45	1.14	0.0284 *
FFMI [kg/m^2^]	0.16	0.48	0.0390 *
BMR [kcal]	14.7	−70.45	0.1167
Albumin [g/dL]	−0.14	0.12	0.0194 *
Transferrin [g/L]	−0.18	−0.16	0.4574
Creatinine [mg/dL]	−0.02	−0.03	0.1666
Urea [mg/dL]	−1.34	0.03	0.4098
GGT [U/L]	14.7	9.76	0.6032
Bilirubin [mg/dL]	−0.21	−0.17	0.3742
Glucose [mg/dL]	−1.42	−2.61	0.6899
TAG [mg/dL]	44.27	61.32	0.1546
HDL Cholesterol [mg/dL]	−5.61	−6.61	0.6495
AlAT [U/L]	5.34	3.26	0.9295
AspAT [U/L]	2.92	2.87	0.5016

Δ—The difference between the final and initial value of the corresponding parameter in the matched control group receiving chemotherapy. Δ_1_—The difference between the final and initial value of the corresponding parameter in the matched supplemented group receiving chemotherapy. Unpaired Student’s *t*-test was used for comparison parameters following the normal distribution. Mann–Whitney U-test was used for comparison parameters not following the normal distribution. * statistically significant.

**Table 4 nutrients-13-03549-t004:** Multivariate regression assessing the influence of age and clinicopathological factors on changes in biochemical and anthropometric parameters during chemotherapy in the matched supplemented group.

Histopathological Grade	Δ BMI	Δ FFM	Δ FFMI	Δ TAG	Δ HDL	Δ Albumine
Odds ratio (OR)	−0.1962	−0.8686	−0.3151	58.70	−2.976	0.2081
95% Confidence interval (CI)	−0.7223–0.3299	−2.415–0.6782	−0.8875–0.2573	−19.06–136.5	−9.595–3.642	−0.3685–0.7846
*p* value	0.4526	0.2609	0.2701	0.1338	0.3661	0.4673
HER-2/neu expression						
Odds ratio (OR)	0.0418	−0.3719	−0.1302	−45.49	1.862	0.3813
95% Confidence interval (CI)	−0.4738–0.5575	−1.888–1.144	−0.6912–0.4309	−121.7–30.72	−4.625–8.348	−0.1838–0.9463
*p* value	0.8697	0.6203	0.6394	0.2326	0.5625	0.1787
Tumor size						
Odds ratio (OR)	−0.0505	0.3440	0.0971	−47.32	−0.6457	−0.4836
95% Confidence interval (CI)	−0.5615–0.4606	−1.159–1.847	−0.4589–0.6531	−122.9–28.21	−7.075–5.783	−1.044–0.0765
*p* value	0.8417	0.6438	0.7241	0.2108	0.8390	0.0881
Regional lymph node metastases						
Odds ratio (OR)	−0.0616	−0.5260	−0.1863	−40.88	1.264	0.2402
95% Confidence interval (CI)	−0.5480–0.4248	−1.956–0.9040	−0.7155–0.3429	−122.8–31.01	−4.855–7.382	−0.2929–0.7732
*p* value	0.7979	0.4588	0.4781	0.2550	0.6765	0.3652
Hormonal sensitivity						
Odds ratio (OR)	0.2628	0.1199	0.0212	21.51	−2.275	−0.3300
95% Confidence interval (CI)	−0.2517–0.7773	−1.393–1.632	−0.5386–0.5809	−54.52–97.55	−8.747–4.197	−0.8938–0.2339
*p* value	0.3056	0.8726	0.9390	0.5681	0.4788	0.2417
Age						
Odds ratio (OR)	−0.0083	−0.0209	−0.0077	−2.395	−0.3546	−0.0010
95% Confidence interval (CI)	−0.03416–0.0176	−0.0970–0.05520	−0.03586–0.02045	−6.219–1.430	−0.6802–0.0290	−0.0294–0.0273
*p* value	0.5190	0.5796	0.5806	0.2111	0.0337 *	0.9411

Odds ratio values and *p* values are indicated, *p* ≤ 0.05 considered as statistically significant. Δ—The difference between the final and initial value of the corresponding parameter in the matched supplemented group receiving chemotherapy. * statistically significant.

**Table 5 nutrients-13-03549-t005:** The influence of initial FM on biochemical parameters in the matched supplemented group after six weeks of chemotherapy.

Analyzed Parameter	FM ≤ 33%			FM > 33%		
	before Chemotherapy	after 6-Week Chemotherapy	P	before Chemotherapy	after 6-Week Chemotherapy	P
Albumin [g/dL]	4.65 ± 0.15	4.70 ± 0.17	0.4159	5.89 ± 7.30	4.42 ± 0.26	0.5905
Transferrin [g/L]	2.93 ± 0.40	2.73 ± 0.40	0.0031 *	2.6 ± 0.36	2.46 ± 0.38	0.0169 *
Creatinine [mg/dL]	0.70 ± 0.11	0.71 ± 0.09	0.6641	0.80 ± 0.09	0.75 ± 0.08	0.0001 *
Urea [mg/dL]	24.88 ± 4.55	27.91 ± 6.79	0.1543	29.47 ± 7.42	28.11 ± 7.48	0.2898
GGT [U/L]	16.25 ± 7.86	21.83 ± 7.26	0.0253 *	26.73 ± 16.17	38.42 ± 33.25	0.0036 *
Bilirubin [mg/dL]	0.43 ± 0.26	0.23 ± 0.08	0.0010 *	0.44 ± 0.20	0.26 ± 0.10	0.0004 *
Glucose [mg/dL]	101.6 ± 10.96	98.50 ± 12.0	0.2760	111.2 ± 25.96	108.8 ± 31.33	0.3964
TAG [mg/dL]	92.17 ± 30.95	121.8 ± 60.51	0.0754	144.5 ± 60.75	225.1 ± 148.1	0.0007 *
HDL-Cholesterol [mg/dL]	68.75 ± 12.35	66.50 ± 11.02	0.3943	56.0 ± 13.82	47.38 ± 11.80	<0.0001 *
AlAT [U/L]	12.75 ± 3.17	15.42 ± 4.87	0.0446 *	20.24 ± 11.02	23.0 ± 11.26	0.0508
AspAT [U/L]	14.58 ± 3.15	16.92 ± 2.47	0.0187 *	19.12 ± 6.53	21.50 ± 6.7	0.0076 *

All results shown as mean ± standard deviations. Paired Student’s test was used for comparison. * statistically significant.

**Table 6 nutrients-13-03549-t006:** Comparison of changes (before and after 6-week chemotherapy) for biochemical parameters depending on initial FM in the matched supplemented group.

Analyzed Parameter	≤33% FM	>33% FM	P
	Δ	Δ_1_	
Albumin [g/dL]	0.05	−1.24	0.5717
Transferrin [g/L]	−0.2	−0.14	0.2926
Creatinine [mg/dL]	0.009	−0.05	0.0047 *
Urea [mg/dL]	3.03	−1.35	0.0626
GGT [U/L]	5.58	11.69	0.4596
Bilirubin [mg/dL]	−0.2	−0.16	0.6601
Glucose [mg/dL]	−3.08	−2.38	0.8892
TAG [mg/dL]	16.1	57.74	0.0258 *
HDL Cholesterol [mg/dL]	−2.25	−8.61	0.0488 *
AlAT [U/L]	2.67	3.54	0.7000
AspAT [U/L]	2.33	3.12	0.8995

Δ—The difference between the final and initial value of the corresponding parameter in the matched supplemented women with normal FM. Δ_1_—The difference between the final and initial value of the corresponding parameter in the matched supplemented women with abnormal FM. Unpaired Student’s *t*-test was used for comparison parameters following the normal distribution. Mann–Whitney U test was used for comparison parameters not following the normal distribution. * statistically significant.

**Table 7 nutrients-13-03549-t007:** The influence of age on body composition and biochemical parameters in the matched supplemented group after six weeks of chemotherapy.

Analyzed Parameter	Age ≤ 56			Age > 56		
	Before Chemotherapy	After 6-Week Chemotherapy	P	Before Chemotherapy	After 6-Week Chemotherapy	P
Albumin [g/dL]	4.52 ± 0.21	4.59 ± 0.24	0.2245	4.45 ± 0.27	4.42 ± 0.27	0.2794
Transferrin [g/L]	2.79 ± 0.48	2.57 ± 0.47	0.0006 *	2.62 ± 0.27	2.52 ± 0.32	0.1281
Creatinine [mg/dL]	0.75 ± 0.12	0.74 ± 0.09	0.1848	0.79 ± 0.08	0.73 ± 0.07	0.0002 *
Urea [mg/dL]	27.21 ± 7.37	26.46 ± 6.31	0.6276	28.92 ± 6.45	29.82 ± 7.82	0.4996
GGT [U/L]	22.30 ± 19.09	36.25 ± 37.49	0.0011 *	24.67 ± 8.19	29.78 ± 14.11	0.1219
Bilirubin [mg/dL]	0.45 ± 0.21	0.23 ± 0.09	0.0002 *	0.42 ± 0.22	0.28 ± 0.10	0.0084 *
Glucose [mg/dL]	101.5 ± 10.12	101.6 ± 17.78	0.5191	115.7 ± 29.77	110.0 ± 34.57	0.0977
TAG [mg/dL]	109.0 ± 62.24	183.5 ± 145.4	0.0007 *	140.9 ± 37.72	169.7 ± 70.04	0.0380 *
HDL-Cholesterol [mg/dL]	60.35 ± 14.26	56.15 ± 14.66	0.0395 *	59.67 ± 15.19	50.39 ± 14.18	0.0008 *
AlAT [U/L]	17.0 ± 12.74	20.0 ± 11.53	0.0201 *	18.67 ± 5.61	21.28 ± 8.96	0.1195
AspAT [U/L]	16.63 ± 6.5	19.3 ± 4.82	0.0071 *	18.72 ± 5.42	20.89 ± 7.26	0.0334 *
Body mass [kg]	74.05 ± 19.42	75.58 ± 20.01	0.0043 *	80.57 ± 13.82	80.91 ± 13.74	0.1877
BMI [kg/m^2^]	26.98 ± 7.37	27.59 ± 7.58	0.0017 *	30.53 ± 4.92	30.62 ± 4.90	0.4274
FM [%]	33.80 ± 8.79	33.51 ± 8.42	0.6947	40.21 ± 6.10	39.19 ± 5.75	0.1261
WHR	0.84 ± 0.09	0.86 ± 0.07	0.2922	0.88 ± 0.06	0.89 ± 0.08	0.4777
Water content [%]	48.74 ± 5.97	49.11 ± 5.84	0.5124	43.98 ± 4.17	42.82 ± 10.44	0.2665
Muscle mass [kg]	45.16 ± 5.43	46.34 ± 6.23	0.0136 *	45.12 ± 5.16	46.12 ± 5.18	0.0487 *
Visceral fat	6.8 ± 3.92	6.78 ± 3.81	0.7455	10.72 ± 2.42	10.53 ± 2.40	0.2650
FFM [kg]	47.58 ± 5.72	48.85 ± 6.52	0.0118 *	47.54 ± 5.43	48.58 ± 5.45	0.0485 *
FFMI [kg/m^2^]	17.43 ± 2.35	17.9 ± 2.67	0.0085 *	17.96 ± 2.11	18.46 ± 2.05	0.0066 *
BMR [kcal]	1640.0 ± 888.4	1475.0 ± 225.5	0.0117 *	1442.0 ± 171.2	1476.0 ± 169.9	0.0028 *

All results shown as mean ±standard deviations. Paired Student’s test was used for comparison. * statistically significant.

**Table 8 nutrients-13-03549-t008:** Comparison of changes (before and after 6-week chemotherapy) for biochemical and anthropometric parameters depending on age in the matched supplemented group.

Analyzed Parameter	Age ≤ 56	Age > 56	P
	Δ	Δ_1_	
Albumin [g/dL]	0.07	0.21	0.2304
Transferrin [g/L]	−0.22	−0.98	0.2924
Creatinine [mg/dL]	−0.01	−0.06	0.2083
Urea [mg/dL]	−0.76	0.9	0.4588
GGT [U/L]	13.95	5.11	0.2412
Bilirubin [mg/dL]	−0.20	−0.14	0.2727
Glucose [mg/dL]	0.15	−5.67	0.2856
TAG [mg/dL]	65.40	56.78	0.3202
HDL cholesterol [mg/dL]	−4.20	−9.28	0.0981
AlAT [U/L]	3.85	2.61	0.5275
AspAT [U/L]	3.50	2.17	0.5083
Body mass [kg]	1.52	0.34	0.0203 *
BMI [kg/m^2^]	0.61	0.09	0.0244 *
FM [%]	−0.29	−1.02	0.3419
WHR	0.01	0.009	0.7570
Water content [%]	0.38	−1.16	0.5986
Muscle mass [kg]	1.19	0.99	0.7672
Visceral fat	−0.03	−0.19	0.3652
FFM [kg]	1.27	1.05	0.7363
FFMI [kg/m^2^]	0.47	0.39	0.7290
BMR [kcal]	−164.8	34.39	0.9069

Δ—The difference between the final and initial value of the corresponding parameter in the matched supplemented women aged 56 and younger. Δ_1_—The difference between the final and initial value of the corresponding parameter in the matched supplemented women older than 56 years. Unpaired Student’s *t*-test was used for comparison parameters following the normal distribution. Mann–Whitney U-test was used for comparison parameters not following the normal distribution. * statistically significant.

## Data Availability

The datasets used and/or analyzed during the present study are available from the corresponding author on reasonable request.

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
