# Peer review of "Beneficial Effects of Oral Nutritional Supplements on Body Composition and Biochemical Parameters in Women with Breast Cancer Undergoing Postoperative Chemotherapy: A Propensity Score Matching Analysis"

_nutrients, 2021, doi:10.3390/nu13103549_

Round 1

Reviewer 1 Report

In the manuscript entitled “Beneficial effects of high-protein oral nutritional support on

body composition, albumin, and lipid level in breast cancer 3 women undergoing postoperative chemotherapy”, the authors described the difference in anthropometric measurements and a group of biomarkers pre- and post-chemotherapy among high-protein supplement takers versus the controls. This is a topic very much worth exploring because of the commonly seen 1) malnutrition status among cancer patients and 2) sarcopenic obesity. This is especially important for breast cancer patients because they tend to have a longer survival time, making them susceptible to comorbidities.

The authors did a great job documenting the study and describing the details. However, I have a few suggestions to perhaps improve the overall strength of the article.

  1. It was not clear how the experiment and control groups were defined. Initially I thought they were randomized, then it seemed that the participants separated themselves according to their personal wills, and finally I saw in the method section the controls were matched (by propensity score or matched on age only?). The authors need to state the classification more carefully because it determines the interpretation of the results.
  2. Page 3, line 97. I did not find the data for physical activity in Table 1.
  3. Page 3, line 100-108. Were all the measurements only taken once before and after, or were there multiple takes to obtain a mean value? How would the authors consider the random errors caused by single measurements?
  4. Page 3, line 105-108. Similarly, how were the samples analyzed in the lab? Were there procedures taken to control any batch effect? These should be documented in methods related to biomarker measurements.
  5. I wonder if the biomarkers under study were picked a priori (meaning that the authors had specific hypothesis for each biomarker to be associated with the supplementation), or only commonly measured lipid profile? If the latter is true, how would the authors consider the possibility of multiple comparisons?
  6. Table 6 and 7. I agree that age might be an effect modifier but why 56 years old? If 56  was a surrogate cutoff for menopausal status, why don’t examine pre/postmenopausal breast cancer patients directly?

Minor suggestions:

  1. Page 2, line 48. Superscript m^2
  2. Page 9, line 233-234. The first sentence doesn’t read well.
  3. Page 9, line 249. Missing a period.
  4. Page 10. This paragraph is too long. Please consider breaking it into a few.

Reviewer 2 Report

Even though the numbers of this study subjects were small and short

duration, high protein supplementation may be a good option to preserve fat free mass or something.

However, continuation of high protein drinking, twice a day may be challenging.

Long period intervention study is needed!

I give you some detailed, small comments to you for revision.

1. Please explain why this study include breast cancer only

2. Please show the compliance to drink high protein supplementation.

3. I wonder overall totoal calorie intake in all subjects. If possible, please show in Table 2. 

Round 2

Reviewer 1 Report

Many questions and concerns were not fully addressed in the revision. Please see in the attachment my point-by-point reply to the authors’ responses.
